# Clinical Applications and Immunological Aspects of Electroporation-Based Therapies

**DOI:** 10.3390/vaccines9070727

**Published:** 2021-07-02

**Authors:** Jean Carlos dos Santos da Luz, Fernanda Antunes, Maria Alejandra Clavijo-Salomon, Emanuela Signori, Nayara Gusmão Tessarollo, Bryan E. Strauss

**Affiliations:** 1Viral Vector Laboratory, Cancer Institute of São Paulo, University of São Paulo, São Paulo 01246-000, Brazil; jc.santosluz@usp.br (J.C.d.S.d.L.); fernanda.antunes@hc.fm.usp.br (F.A.); nayara.tessarollo@hc.fm.usp.br (N.G.T.); 2Laboratory of Experimental Oncology, Cancer Institute of São Paulo, University of São Paulo, São Paulo 01246-000, Brazil; mariasalomon@usp.br; 3Institute of Translational Pharmacology, CNR, 00133 Rome, Italy; emanuela.signori@cnr.it

**Keywords:** bleomycin, electrochemotherapy, electroporation, gene electrotransfer, vaccines

## Abstract

Reversible electropermeabilization (RE) is an ultrastructural phenomenon that transiently increases the permeability of the cell membrane upon application of electrical pulses. The technique was described in 1972 by Neumann and Rosenheck and is currently used in a variety of applications, from medicine to food processing. In oncology, RE is applied for the intracellular transport of chemotherapeutic drugs as well as the delivery of genetic material in gene therapies and vaccinations. This review summarizes the physical changes of the membrane, the particularities of bleomycin, and the immunological aspects involved in electrochemotherapy and gene electrotransfer, two important EP-based cancer therapies in human and veterinary oncology.

## 1. Introduction

The permeabilization of the cell membrane by the application of external electric pulses was first described in 1972 by Neumann and Rosenheck [1,2]. This physical process was coined as “electroporation” in 1982 by the same group [3]. Such permeabilization is characterized by the uptake of external nonpermeant molecules in the cell cytoplasm and leakage of internal substances from the electropermeabilized cells [4,5,6]. Since then, this technique has being used and developed in several areas of biotechnology and medical research, such as bacterial transformation [7,8], cell electrofusion [9,10] transdermal drug delivery [11,12], CRISPR editing [13], CAR-T cell generation [14], DNA vaccines, including for the novel Sars-CoV-2 coronavirus [15,16,17], gene therapy [18,19,20,21], cancer treatment [22,23,24] and even in food processing [25,26] and environmental sanitation [27,28].

Although electroporation (EP) and electropermeabilization are frequently used as synonyms, electroporation is related strictly to the aqueous pores formed in the lipid membranes of the cells, while electropermeabilization is related to all the events involved with the membrane permeabilization process, including modulation of membrane channels, and cellular biophysical and biochemical changes [5,29,30,31]. There are two main classifications of electropermeabilization, irreversible electropermeabilization (IRE), and reversible electropermeabilization (RE), which are related to different effects on cell membranes, depending on the electrical parameters employed [4,6]. IRE is a tissue ablative therapy, causing cell death mainly by apoptosis and necrosis without drug intervention. The irreparable disruption of plasma membrane and, subsequently, the influx and efflux of soluble molecules leads to loss of cell homeostasis [23,32,33]. This technique was introduced as cancer therapy by Dr. Rubinsky’s group in 2005 [34]. Moreover, another IRE technique, named nanosecond pulsed electric field (nsPEF), has also been used in oncology. This approach uses electric pulses to create irreversible pores in plasmatic and also intracellular membranes, leading to cell death [35,36]. In turn, RE is used in oncology for electrochemotherapy (ECT), gene electrotransfer (GET or, the less used term, electrogenetherapy—EGT) and electrochemogenetherapy (ECGT), aiming to permeabilize the cells to allow the entrance of external therapeutic molecules, but without irreversible damage to the cell [4,35,37].

This review focuses on the physical changes of the membrane, the particularities of bleomycin, and the immunological aspects involved in EP-based therapies and their application for the treatment of cancer both in humans and animals.

## 2. Plasma Membrane Electroporation

The phospholipidic components of the plasma membrane possess a polar hydrophilic “head” and a nonpolar hydrophobic “tail”. In this amphiphilic structure, the phospholipids are spontaneously organized in the aqueous physiologic environment, arranging their hydrophilic heads together in the outer region of the membrane and aggregating hydrophobic tails in the inner part, forming a stable, bilayered closed vesicle [38,39]. The plasma membrane acts as a regulatory barrier, separating the cell interior from the exterior. This structure is permeable to water, oxygen (O_2_) and carbon dioxide (CO_2_), relatively permeable to hydrophobic molecules and poorly permeable to hydrophilic molecules, allowing only membrane-controlled ions and molecules to cross in and out of the cell through diffusion channels, and membrane transport proteins [33,39,40,41]. Thus, cells can naturally maintain an electric potential difference between the internal and external side of the plasma membrane, the so-called resting transmembrane potential [29,41].

Cell membrane EP requires well-defined electric pulse protocols, regarding the number, duration, voltage amplitude, frequency and electric current applied to the tissue depending on its intended use. These electric pulses are generated by specialized equipment and delivered to the cells or tissue through electrodes (commonly needle or plate electrodes) with positive and negative poles, with the electric field occurring between those poles, positioned around the site of interest [30,42,43,44]. While in vivo RE uses electric parameters varying around eight square pulses of milli- to microseconds of duration and 100–1000 V of amplitude and only causes transient permeability with subsequent membrane resealing, in vivo IRE uses more pulses (usually around 80–100 pulses), decreased pulse durations and higher voltages (around 3000 V), leading to cell death due to incapacity of cells to recover [4,6,35,36]. Hypothetically, it is possible to achieve EP in all cell types, in vitro and in vivo, as long as its particular characteristics are respected, such as physical (e.g., cell and electrode sizes and shapes), biological (e.g., cytoskeleton structure), tissue localization and adequate electric parameters [45,46,47].

When an external electric pulse exceeds a threshold value, it creates an electric field that can modify the resting transmembrane voltage (RTV), leading to an induced transmembrane voltage (TMV), causing structural and molecular changes to the plasma membrane components [5,29,48]. Below the threshold of TMV, no detectable EP occurs. With an adequate electric field, the TMV threshold is achieved, enabling an RE during the pulse application. In the presence of stronger electric fields, a nonthermal IRE takes place, which leads to cell death, and may be used as a cancer treatment. Otherwise, with even stronger electric fields, thermal IRE takes place, leading to thermal damages to the tissue, and consequently cell death. This approach is not used in cancer treatment [49,50].

Kotnik and collaborators [29] reviewed the RE process and divided it in five phases—Initiation: changes in plasma membrane electrical (conductivity) characteristics in the first nanoseconds and increase in its permeability in the following microseconds, when TMV exceeds a threshold value; Expansion: with the persistence of TMV above the threshold value, augmented conductivity and permeability of plasma membrane remains and/or intensifies until the pulse ceases; Partial recovery: with the pulse cessation, the TMV returns below the threshold value and membrane conductivity and permeability decrease partially (microsecond and millisecond, respectively), remaining stable with ions and small molecule diffusion; *Resealing*: the gradual recovery of the membrane homeostatic impermeability in RE of cells (and nonrecovery in case of IRE), taking seconds to minutes at temperatures around ~20–37 °C, and up to hours at temperatures around ~4 °C; and, finally, Memory: normal membrane permeability with cellular physiological alterations and reactions to stressors during the first hours post pulse cessation; after this period cell recovers its homeostatic state [29,48].

From a physical perspective, the pore formation can be divided into: (A) normal non-electroporated membrane; (B) early formation of unstable water channels; (C) metastable pore formation (reversible electroporation); and (D) pore destabilization (irreversible electroporation) [29,43,51] as demonstrated in Figure 1.

Beyond pore formation, modulation (opening) of voltage-gated channels and clustering of transmembrane proteins (creating temporary pseudotunnels) also contribute to the crossflow of physiological and inoculated substances [12,29,31].

## 3. Clinical Aspects of EP-Related Therapies

### 3.1. Bleomycin

Bleomycin and cisplatin are the most widely used chemotherapeutic drugs in electrochemotherapy. Both present poor cell membrane penetration, and their cytotoxic potential is increased up to 8000 times and up to 80 times, respectively, when combined with EP [24,52]. Furthermore, bleomycin, the drug highlighted in this review, has other important features: it has minor side effects at the recommended doses for EP-therapies, presents selectivity for tumor cells, reduces tumor oxygen and nutrients supply by its antivascular effects and can be used intratumorally and systemically (intravenously) in humans, dogs, and cats [53,54,55,56,57,58].

The cytotoxic potential of bleomycin relies on its ability to cause single- and double-stranded DNA breaks, acting as an endonuclease [58,59], and alternatively also cleaves RNA molecules such as mRNA, tRNA and rRNAs [60]. Inside the cell, bleomycin can cause two different types of cell death, depending on the number of molecules internalized by the tumor cell [59,61]. The level of internalization is related to the route used to administer the drug, that is, intratumoral (IT) or intravenous (IV) systemic administration. When administered intravenously, the interstitial concentration of bleomycin at the time of application of the electrical pulses is low, thus allowing the establishment of mitotic cell death; in cases of IT application, the internalized concentration is higher, leading to the so-called pseudoapoptotic cell death [58,59,61,62].

The antitumor selectivity of bleomycin resides in the mitotic cell death (i.e., intravenous route), characterized by the appearance of TUNEL and caspase-3 positive cells and atypical mitotic cells. When low numbers of molecules are internalized, nonlethal damage occurs in the DNA. Such damage is shown to be cytotoxic only at the moment of cell division, since the cells’ ability to repair the DNA damage is impaired; the cells in constant replication (such as tumor cells) are led to cell death by apoptosis, while nondividing healthy cells around the treated tumor are spared, as the chances of such DNA damage affecting the expression of vital genes is almost null. This characteristic is particularly important in cases of narrow safety margin resection, allowing a “cleaning” of possible residual tumor cells in healthy tissues, when this approach is used as adjuvant to surgeries [31,58,59,61,63]. In the case of intratumoral administration, pseudoapoptosis predominantly takes place, characterized by a rapid appearance of apoptotic cells due to the high number of bleomycin-induced breaks in the double-strand DNA. In this case, the characteristic of tumor cell selectivity is diminished due to the numerous breaks of DNA strands, acquiring a cytotoxic profile even in the absence of cell division, causing cell death in the treated area [58,61,62].

Recommended doses of bleomycin for human treatment in intravenous bolus injection is of 15,000 IU/m^2^ (patient’s body area) and intratumoral dose according to tumor nodule size: bleomycin dose of 250 IU/cm^3^ for nodules >1 cm^3^, bleomycin dose of 500 IU/cm^3^ for nodules >0.5 cm^3^ and smaller than 1 cm^3^ and 1000 IU/cm^3^ for nodules smaller than 0.5 cm^3^ [24]. For dogs and cats, the recommended dose in intravenous bolus is 300 IU/kg of patient weight and intratumoral dose according to nodule size: 1500–3000 IU/cm^3^ for tumor nodules >1 cm^3^ and 1500 IU/cm^3^ for tumor nodules smaller than 1 cm^3^ [54]. A benefit of the systemic injection of bleomycin lies in the possibility of treating multiple nodules or large nodules after a single drug injection, avoiding repetitive bleomycin application and its toxic cumulative dose of 400,000 IU/m^2^ in human, dogs, and cats [54,64].

### 3.2. Electrochemotherapy

The term electrochemotherapy (ECT) was coined by Mir et al. in 1991 in one of the first studies to introduce the use of ECT, where they described the potentiation of bleomycin’s cytotoxic effects in murine models [65]. The first human clinical trial using ECT was also performed in 1991 [66] and the first veterinary clinical trial in 1997 [67]. Since then, ECT evolved from a rescue to a primary option for the treatment of cancer, both in humans and animals. This approach gained prominence in 2006, mainly due to the European Standard Operating Procedures of Electrochemotherapy (ESOPE), a highly successful multi-institutional effort that aimed to establish standard operating procedures for ECT in humans. Currently, ECT is increasingly being introduced into clinical guidelines and evolving with multi-institutional collaborative research groups (e.g., InspECT, EURECA, Gisel, and ReinBone) and new technologies and procedures [24,42,68,69], with important contributions from veterinary patients in these developments [70,71].

Nowadays, ECT has been included in several guidelines from many countries for the treatment of different cancer types, including melanoma, breast cancers, head and neck cancers, carcinomas, soft tissue sarcomas and bone metastases [42,72]. Among the different indications of ECT for cutaneous and subcutaneous primary cancer or skin metastasis, the following are included: palliative treatment of skin metastases in advanced diseases; neoadjuvant cytoreductive therapy prior to conventional treatment; organ and function sparing treatment; patients not suitable for or relapsed after treatment by surgery, radiotherapy or systemic therapy, and treatment of highly vascularized nodules due to its antivascular effect [42,73,74,75,76]. Additionally, in veterinary medicine ECT is used as an adjuvant therapy for the treatment of incompletely resected tumors, aiming to remove possible residual infiltrating cancer cells at the surgical site, when a wide safety margin resection is not possible [31,77,78]. The use of ECT in noncutaneous deep-seated and internal organ tumors is under investigation in a number of human clinical trials, with preliminary results indicating its feasibility and safety [42,72,79,80,81,82,83]. Currently, novel approaches using ECT are under investigation in pancreas, bone, liver, bladder, prostate and gastrointestinal tumors; other studies indicate the potential of ECT in lung and brain tumors, suggesting that new future approaches might be translated between human and veterinary care [68,70,79,84,85].

Technically, ECT can treat any type of solid tumor as long as it is accessible to electrodes, whether cutaneous or deep-seated, and accessible through endoscopy, laparoscopy or open surgery [64], but it may present different levels of effectiveness in different tumor types, possibly due to insufficient drug and/or electric field coverage and also due to patient biological features, such as previous treatments, tumor size, intrinsic tumor cell sensitivity to the drugs, drug distribution through tumor vascularization, tumor immunogenicity and patient immune status [72,86,87,88,89,90]. This approach has numerous advantages when compared to other locoregional antitumor therapies, such as ease of administration, low cost, minimal side-effects (generally limited to mild local inflammation and transitory pain in the treated site), possibility of repetition if necessary, and high response rate, consistently obtaining approximately 80% objective response (OR) rates in all tumor histologies and around 60–70% CR (complete response) rates after a single treatment in humans [42,53,70,76,87]. As an example, in a study of 52 patients with cutaneous or subcutaneous metastasis of different tumor types, Campana and colleagues (2009) obtained approximately 96% (50 of 52) OR (RECIST) one month after the treatment with ECT plus bleomycin. Specifically, 50% (26 of 52) of those patients had a CR and 46% (24 of 52—RECIST) had partial response (PR); there was no change in tumor size in the other two (4%—RECIST) patients after treatment. Between the partial responders and nonresponders, an improved response was obtained after, at least, one additional ECT session. Among the 26 complete responders, 17 were disease-free at the treated site after a mean follow-up time of 9 months. However, 7 of the 26 complete responders developed new lesions in untreated areas, one presented local recurrence after seven months and another two patients died from systemic disease progression [91], indicating a localized ECT efficacy in those patients, though further studies with larger cohorts are warranted. Similarly, Campana and colleagues (2016), in a prospective analysis of 376 patients with different tumors, obtained CR above 80% (RECIST) regardless of tumor histotypes, treated by single or multiple sessions of ECT plus bleomycin, however, no observation of regression in untreated tumors, reinforcing the strictly local effectiveness characteristic of the technique [92].

Similar to surgical approaches as a local treatment, an adequate application of ECT must uniformly cover the whole tumor tissue and its surrounding safety margins with an appropriate electric field and sufficient drug at the moment of the delivery of electric pulses [42,89]. To accomplish this, the pulses must be applied between 1 to 10 min after intratumoral bleomycin application and between 8 to 28 min if intravenously applied, consistent with the period of bleomycin’s pharmacokinetic peak [24,93,94]. After drug administration, the electrodes must be systematically moved and the electrical pulses applied in the new site (Figure 2), treating the entire tumor area [42].

Indeed, the efficiency of EP in RE therapies depends on several factors, such as the electrical parameters, tumor location and composition of the extracellular matrix and the type and dose of plasmid administered [95]. Moreover, different types of electrodes can be used to deliver genes: needle electrodes [96], microneedles [97], plate electrodes [98] and noninvasive multi-electrode array (MEA) [99]. In general, plate electrodes are used for superficial tumors and needle electrodes are used for deeper tumors [42]. Internalization of plasmid vectors represents a crucial point for GET therapies and further studies aiming to reach a major plasmid intake and expression are mandatory [100,101]. One strategy is the employment of hyaluronidase enzyme. This enzyme disrupts the extracellular matrix, allowing the DNA to reach the cell membrane more easily, where it will be internalized by the cells when pulses are administered [102,103].

The voltage is also important for drug delivery. Typically, small molecules (e.g., bleomycin and cisplatin) require short high-voltage pulses, while large molecules need pulses of longer duration that are either low-voltage or a combination of high- and low-voltage. The electric field is distributed according to the electrode geometry and tissue conductivity [104], and the cells that first come into contact with the proper electric field will be the first to be electropermeabilized. [46,105]. Thus, electrical dosage in tumors depends on the tumor’s conductivity, volume, position and also electrode configuration [106]. Recent papers have reported the increasing importance of the target tissue in the efficacy of gene electrotransfer, which electrical parameters should be determined for each tissue type, such as liver, skin, muscle or tumor, due intrinsic differences between them [107].

## 4. Immunological Aspects of Reversible Electroporation and Electrochemotherapy

While ECT is recognized for its cytotoxic clinical benefit, the immunological potential of RE alone and ECT deserve particular attention. As described here, these effects play an important role in promoting antitumor immune responses.

According to the “danger theory”, introduced by Matzinger in 1994, apoptotic and necrotic cells could act as immunostimulants by releasing damage-associated molecular patterns (DAMPs) [108,109,110] that trigger proinflammatory responses. This type of cell death is known as immunogenic cell death (ICD) [111]. The damage caused to the tissue by EP-related therapies can elicit ICD and promote a proinflammatory milieu and the recruitment of antigen presenting cells (APC) where tumor-specific antigens (TSA) and tumor associated antigens (TAA) might be captured and presented by APCs to T cells, leading to adaptive immune responses [112,113,114,115,116,117]. In fact, RE alone or in combination with bleomycin can lead to the leakage of cytoplasmatic molecules related to ICD and the activation of proinflammatory markers [114,118,119,120]. The main immunostimulatory molecules related to RE include: adenosine triphosphate (ATP) release, calreticulin (CRT) translocation to cell membrane, release of the non-histone, protein high mobility group box 1 (HMGB1), induction of heat shock protein family (HSP70) stress response and other proinflammatory genes [114,121,122]. These markers contribute to the induction of the host’s adaptive immune response.

ATP, a major immunostimulatory DAMP related to ICD, promotes the recruitment, maturation and cross-presentation by APCs [114,121] supporting immune responses in EP-related therapies. Calvet and colleagues (2016) demonstrated that RE alone was capable of releasing ATP from CT26 murine colon cancer cells immediately after the electric pulses and lasting up to hours. However, no ATP was detected 30 h after the electric pulses alone. In contrast, a significant amount of ATP was detected 30 h after EP plus bleomycin, even in cells that were washed to removed ATP released by the electric pulses, indicating a continued leakage of ATP even after the cell membranes completely reseal, and pointing to the mandatory presence of bleomycin for the continued release of ATP at the treated site [114].

CRT promotes the secretion of type I interferons and the uptake of dying cells by APCs [121]. Two possible mechanisms for CRT release are: (i) direct ER stress caused by the electric pulses and/or (ii) the generation of reactive oxygen species (ROS), which causes ER stress and indirectly leads to CRT exposure on the cell surface [114,123,124]. Schultheis and collaborators (2018) evaluated the effects of electric pulses alone on tissues using RE in the Hartley guinea pig model. They showed that CRT translocation to cell membrane was upregulated after RE alone, corroborating a previous study using ECT plus bleomycin in vitro [114,119], and CRT was detected along the entire area of the electric field, returning to normal levels after 24 h post-RE. Furthermore, infiltration of APCs such as dendritic cells (DCs) and macrophages at the treatment site was also reported, with its peak level coinciding with the peak of CRT translocation to the cell membrane [119]. In fact, melanoma patients exhibited CRT+ cells after treatment with ECT plus bleomycin as detected by fluorescent immunochemistry up to 14 days later, and this longer duration of CRT exposure may be related to the internalization of bleomycin by the cells [114,125].

HMGB1 is a protein that stimulates the synthesis of several proinflammatory cytokines, such as TNF, IL-1α, IL-1β, IL-6, IL-8, MIP-1α, MIP-1β and is able to drive the maturation and cross-presentation of antigens by APCs [121,126]. Calvet et al. (2014) showed that RE alone did not trigger the release of HMGB1 in CT26 tumors in vivo, possibly due to its dependence on the cell death processes. However, in the presence of bleomycin alone or in combination with electric pulses (ECT), cell viability decreased with consequent HMGB1 release, where ECT was associated with a higher level of HMGB1 release than bleomycin alone [114].

Gene expression profiles have been shown to be altered after RE. In SK-MEL28 melanoma cells exposed to RE, differences were revealed in seven genes related to HSP70, which is recognized for promoting the uptake of dying cells by APCs, repression of H4 histone and protein synthesis downregulation [121,122]. Moreover, no changes of major tumor suppressor genes, oncogenes or genes involved with DNA stability were observed, supporting the notion that EP does not induce tumorigenesis [122]. On the other hand, RE caused the activation of proinflammatory chemokine genes, including the expression of MIP-1α (CCL3), MIP-1β (CCL4), MIP-1γ, IP-10, MCP-2, and XCL1, which trigger the influx of inflammatory APCs to the treated site [127].

Moreover, EP can also induce a bystander effect, seen when using conditioned medium, possibly through the release of microvesicles, with a decrease of viability in the cell lines exposed to medium from RE and IRE treated cells. The number of microvesicles and the induction of cell death upon exposure to the conditioned medium are increased as the electric field intensity and the number of applied pulses rise, and also varies according to cell type [128].

Regarding immune status and ECT responses, Sersa and colleagues (1997) compared the outcomes in C57BL/6 and Swiss nude mice after treatment with ECT plus cisplatin aiming to evaluate the antitumoral response of immunocompetent and immunodeficient mice [129]. Nude mice are characterized by an absence of thymus that consequently leads to a lack of T cells (both CD4+ and CD8+) while keeping intact innate populations such as natural killer (NK) cells [130]. In their experiment, up to 82% (14 of 17) of immunocompetent mice bearing LPB sarcomas were cured by ECT plus cisplatin, whereas no therapeutic response was observed in nude mice. Furthermore, after a rechallenge with LPB tumor cells, 75% of the cured mice previously treated by ECT rejected the tumor, whereas no rejection was observed in the control group [129]. A similar outcome was described by Mir and colleagues (1991) using ECT plus bleomycin, where immunocompetent mice demonstrated better outcomes when compared to immunodeficient nude mice [65]. Both studies confirm the importance of host immune responses, especially T-lymphocytes, in the cure rate of the treatment.

However, the immunological response seems to be limited. For example, melanoma patients exhibited increased tumor infiltration by CD3+ CD8+ and CD3+ CD4+ and decreased CD4+ FOXP3+ Tregs after treatment with ECT plus bleomycin. Among the treated patients, 60% (6 of 10) demonstrated CR of treated lesions and 40% (4 of 10) patients displayed PR. Even so, four patients presented new lesions despite CR or PR after the previous treatment. In addition, only a low level of calreticulin exposition was observed on the first day post-treatment, followed by even weaker exposition at day 14 post-treatment [125].

Together, these findings indicate that despite the induction of ICD and local proinflammatory milieu, ECT by itself is not enough to induce antitumor effects on distant untreated lesions but retains an intrinsic potential for systemic and long-lasting effects when combined with immunostimulants [63,113,131,132,133,134]. In this way, the rationale is that the proper activation of APCs and other innate cells within the tumor microenvironment upon EP-related therapies ultimately must lead to the priming of adaptive immune cells, that not only contribute to the killing of tumor cells, but also lead to the establishment of immunological memory that will protect in case of tumor relapse or metastasis. To this end, two different approaches can be mentioned, gene electrotransfer and electrochemogenetherapy, both of which aim to boost the local immunological response by taking advantage of the delivery system provided by the electric pulses.

## 5. Gene Electrotransfer (GET) and Electrogenechemotherapy (EGCT) Alone or in Combination with Other Immunotherapies

Due to recent advances in understanding the biology of the immune system, new strategies such as those based on gene transfer have been proposed to treat infectious and cancer diseases. Genetic immunization transfers into the host a gene sequence encoding antigens or immunomodulatory molecules in order to induce an immune response. To date, the vast majority of gene therapy clinical trials have addressed cancer (67.4%), monogenic diseases (11.6%), cardiovascular and infectious diseases (5.8%) [135,136]. Among the nonviral vectors, plasmid DNA represents a good option for gene therapy purposes due to its safety, flexibility in design, cost-effective production in large scale, and stability at different temperatures. On the other hand, plasmid DNA itself is poorly immunogenic, and its introduction into host cells may require assistance by chemical or physical approaches [137,138].

Electroporation emerged as a suitable strategy for introducing plasmid DNA into target cells, paving the way for gene therapy mediated by electroporation, namely, GET, to be recognized as a useful approach for treating infectious and cancer diseases. Due to the perturbation of the cell membrane induced by the administered pulses, EP has been demonstrated to yield a higher number of transfected cells and larger quantities of plasmid permeating into each cell [139,140]. Furthermore, the local tissue damage induced by EP has been shown to play a role in enhancing both arms of the immune response, thus favoring immunization protocols [141]. GET-based preclinical protocols have been tested in different animal species (mice, rats, rabbits, and pets such as cats, dogs, horses) and tissue-specific protocols have been set up in skin, muscle, tumor, liver, cornea, lung, kidney, brain, bladder, and testis. These studies have led to the establishment of different delivery conditions making efficient DNA electrotransfer feasible while minimizing possible electrical damage and maintaining effective protein expression [4,142,143].

Currently, human (Table 1) [144] and veterinary [145] clinical trials have been approved for antitumor therapies based on EP delivery of antitumoral drugs (bleomycin) or plasmids (cytokines and neoantigen vaccines) combined or not, are ongoing, mainly in the USA and Europe. The most advanced anticancer immune therapy based on plasmid DNA and electroporation is the GET of interleukin-12 (IL-12), which is currently on fast-track in the USA for orphan drug status [146].

IL-12, the most widely used cytokine in GET and EGCT protocols, connects the innate and adaptive immune responses, and presents inherent antitumor activities [142,147,148]. IL-12 is released by APCs after activation by pathogen-associated molecular patterns (PAMPs) and DAMPs and triggers the secretion of interferon-γ (IFN-γ) by T cells and NK cells. This cytokine is a key signal of cross-presenting by DCs to naive CD8+ T cells during priming and polarizes T cells towards the Th1 and Tc1 cytotoxic profile, promoting their conversion into IFN-γ producer cells and shifting myeloid-derived suppressor cells to functional APCs [149,150,151,152,153,154]. In turn, IFN-γ may also act directly on tumor cells, upregulating the expression of HLA class I and II, enhancing their recognition by CD8+ T cells through MHC class I processing and presentation of TSA and TAA [147,155,156]. In other words, IL-12 can transform noninflamed tumors into inflamed Th1-polarized tumors [149]. B16F10 tumors treated with IT IL-12 GET demonstrated a rapid induction of IL-12 expression, increase in T cell infiltration (TILs), upregulation of antigen-presenting genes, recruitment and polarization of antitumoral M1 macrophages, decrease of PD-1 expression and induction of memory T cells when compared to control groups [142,157,158]. Interestingly, it was observed that distant untreated lesions also had increased TILs, particularly of KLRG1^hi^ CD8+ effector T cells, a population that can differentiate into a memory T cell subset capable of mounting highly effective anti-tumor responses with improved resistance to T cell exhaustion by PD-1 and CTLA-4 immune checkpoints [142,159].

Mechanistically, IT delivery of IL-12 plasmid applied once in B16F10 cells resulted in 70% (14 of 20) of tumor remission in treated animals, with significant therapeutic effects against untreated tumors; the lack of perforin or IFN-γ and the blockage of CD8+ T cells abolished IL-12 GET efficacy, whereas the blockage of CD4+ T cells or NK cells did not. IL-12 GET induced a specific CD8+ T cell population (against epitope Trp2180-188) with approximately 33.3% (2 out of 6) of the cured animals being protected against tumor rechallenge. Concurrent Trp2 peptide vaccination increased protection and in turn of 85.7% (6 of 7) cured mice were able to reject tumors [160]. As a result, preclinical animal models treated by EP delivery of IL-12 demonstrated tumor regression, long term survival and resistance to rechallenge [99,158,161,162]. Furthermore, in veterinary oncology, administration of IL-12 GET alone or in combination with ECT has been performed in client-owned dogs with spontaneous tumors, observing enhanced antitumor effectiveness and safety, opening new perspectives for human treatments [163,164,165].

Aiming to synergistically join the antitumoral activities of ECT and GET, some studies proposed ECGT, uniting the local ICD effect of ECT with the enhanced proinflammatory milieu of IL-12 (or other proinflammatory proteins) GET, which may lead to memory and systemic (abscopal) immune response even in untreated lesions. In this approach, the tumor antigens released by the treatment act like an autologous vaccine [63,132]. The treatment of B16F10 tumors with ECGT of IT IL-12 GET plus bleomycin led to complete remission in 37.5% (3 of 8) of treated mice by 150 days after treatment, with 100% (3 of 3) of those establishing tumor resistance upon rechallenge [166]. In canine mast cell tumors treated by ECGT, a pronounced and long-term antitumor effect was observed in patients treated by single or multiple peritumoral (PT) IL-12 GET with intratumoral cisplatin or bleomycin, with 72% (13 of 18) CR (RECIST) and prevented local recurrences and distant metastases (median 40 months of follow-up) [163]. Milevoj et al. obtained a 67% (6 of 9) objective response rate (RECIST) in canine patients with oral melanoma treated by cytoreductive surgery plus ECGT of PT IL-12 plus bleomycin in the tumor site. A decline in peripheral Treg cells was reported after the treatment, with no changes in peripheral CD4+ and CD8+ T cell populations [167]. In another study, a clinical response to ECGT of IT IL-12 plus bleomycin was obtained in dogs with different types of spontaneous tumors. CR was obtained in 50% (3 of 6) and the other three had PR to the treatment [165].

Other proposed approaches include the combination of EP-related therapies with immune checkpoint blockade (ICB) or DNA vaccination. It is known that systemic immunotherapy options such as vaccination and ICB have shown limited antitumoral responses due to the lack of tumor antigen availability, poor antitumoral immune cell infiltration and tumor immunosuppressive microenvironment [168,169], representing possible targets for the combined therapy. In a case report of metastatic melanoma treated by ECT and subsequent administration of ipilimumab (a CTLA-4 inhibitor), Brizio and colleagues (2015) obtained CR of all lesions, even in the untreated internal metastasis, in an abscopal manner [170]. Mozzillo and colleagues (2015) conducted a retrospective analysis of advanced melanoma patients treated with ECT plus ipilimumab [171], concluding that the combination seemed to be beneficial to the patients, with reasonable responses obtained. They also indicated that circulating Treg levels might be a potential predictive biomarker of tumor response for this combined approach [171]. In another case report, Karaca and colleagues (2018) treated a patient with metastatic melanoma with ECT plus nivolumab (a PD-1 inhibitor). Although the patient had previously been treated with different therapies, including chemotherapy, targeted therapy, and immunotherapy, with no consistent tumor response, yet after the combined therapy they obtained a CR with no evidence of cutaneous or internal lesion during 4 years of follow-up [172]. Furthermore, ECT and CTLA-4 and PD1 inhibitors have been evaluated in a retrospective study of melanoma patients. The authors found that ECT plus PD-1 inhibitors had better outcomes than the combination with CTLA-1 in terms of OR (19.2% and 40%, respectively—RECIST) [173].

In a recent phase II clinical trial, patients with nonimmune-infiltrated “cold” melanoma were treated by a combination of IT IL-12 GET plus pembrolizumab (an anti-PD-1 checkpoint inhibitor). Patients had a 41% (9 of 22—RECIST) overall response rate (ORR), with 36% (8 of 22—RECIST) of CR, and exhibited significant local and systemic antitumor response, including those with previous unsuccessful anti-PD-1 therapy, with no new or unexpected toxicities related to pembrolizumab. Upregulation of innate and adaptive immune-associated genes and proinflammatory cytokines and chemokines expression was observed, and enhanced T cell infiltration in treated and untreated lesions and an increase in CD8+ effector memory T cells among peripheral blood was reported [149]. Recently, using a murine model of melanoma, intradermal IL-12 GET plus DNA vaccination combined with anti-CTLA4/PD-1 blockade led to tumor reduction through long-lasting antigen-specific IgG antibodies together with antigen-specific CD8+ T cells [174]. In another approach, using EP as a delivery method for the intradermal Her2/neu DNA vaccine in murine models, Lamolinara and colleagues (2015) obtained a cytotoxic response in tolerant mice and a humoral (antibody) response in both tolerant and nontolerant mice [175].

## 6. Variability of Responses in EP-Related Therapies

The different outcomes in EP therapies demonstrated in the literature stem from biological factors, such as the individual, tumor types, intrinsic variability and the diverse protocols used in studies and clinical trials, which include different application routes of drugs (intravenous or intratumoral) and plasmids (intratumoral, peritumoral and intramuscular), different doses and types of therapeutic molecules, as well the diverse electrode designs. In addition, much of the data has come from early phase clinical trials performed in a limited number of patients which may add to the observed variability. In a systematic review of local effectiveness of single-session ECT comprising 413 patients with different tumor types and 1894 nodules (evaluated by WHO or RECIST criteria), the authors showed a higher antitumor response of IT treatment (72.7% of CR and 85.8% of OR) when compared to the IV administration of bleomycin (54.9% of CR and 80.7% of OR), but with no significant difference when comparing tumor response to IT administration of bleomycin and IT cisplatin. Further, they did not observe different outcomes in human patients with different tumor types (melanoma, carcinoma and sarcoma) treated by ECT [176].

The inability to generate a systemic immune response and the regression of untreated lesions in the form of an abscopal effect, even with the modulation of the local microenvironment in the ECT-treated lesions, may be related to previously established immunosuppression in these untreated lesions and intrinsic characteristics of the tumors [133,177], which may also occur in other EP-related therapies. For example, it was shown that a higher number of CD8+ cells infiltrating tumors prior to ECT treatment was associated with better local response (7 responders of 9 treated—WHO criteria) when compared to the low presence of those cells (1 responder of 6 treated). Interestingly, the infiltration of FOXP3+ cells was not associated with ECT local response, however, it was significantly associated to the development of visceral metastasis [178], corroborating similar results obtained by Ursic and colleagues (2021) in three different tumor models treated by ECT, where the more immunogenic CT26 tumors had better outcomes when compared with less immunogenic 4T1 and B16F10 tumors [179]. In addition, in a comparative study of a murine model of sarcoma (immunogenic) and carcinoma (less immunogenic) treated by ECGT of IT cisplatin plus single or multiple IM IL-12 GET, it was demonstrated that the antitumor effect depends in part on the immunogenicity of the treated tumor, indicating that more immunogenic tumors had a better response and higher cure rate [90]. Those findings reflect the role of individual and tumor immunologic profiles in the outcomes of EP-therapies.

Regarding the route of plasmid administration, it was shown that IT IL-12 GET led to NK and CD8+ and CD4+ T cell accumulation and tumor elimination in murine squamous cell carcinoma. However, the antitumor response through the IM route was mediated mainly by NK cells, with minimal activation of specific CD8+ T cells and, consequently, inefficacy in tumor elimination [162]. A similar response was observed by Lucas and colleagues (2002) comparing both administration routes. The authors found that IT injection promoted the cure of 47% (8 of 17) of B16F10 melanoma tumors, prolonged survival and resistance against tumor rechallenge in 71.4% (5 of 7) of the cured animals, whereas the IM treatment did not result in tumor regression. Furthermore, they found increased levels of IL-12 and IFN-γ, and influx of CD4+ and CD8+ T cells into the treated tumors. In addition, both IT or IM treatment did not regress B16F10 tumors in the nude mouse group, reinforcing the role of lymphocytes in tumor elimination [180].

In another cancer model, murine subcutaneous fibrosarcoma treated by IT or PT IL-12 GET led to CR in 90% (18 of 20) of the IT treated mice and 15.6% (3/19) in the PT group; among cured mice, 61.1% (11 of 18) from the intratumoral group and 100% (3 out of 3) from the peritumoral group were protected against tumor rechallenge. The treatment in both regions promoted increase of IL-12 and IFN-γ level in serum and into the tumor and delayed the tumor growth of untreated lesions [181]. In humans, some evidence of complete response was observed in patients with Merkel Cell Carcinoma (MCC) treated with intratumoral IL-12 GET; 33.3% (1 of 3—RECIST) patients with local MCC showed pathological complete remission while 16% (2 of 12—RECIST) patients with metastatic MCC experienced prolonged survival. The responses were associated with the increase of peripheral and intratumoral MCC-specific T cells, also present in non-treated lesions, indicating an abscopal effect [182].

Indeed, even the level and duration of protein expression can be modulated to some extent by different plasmids, electrodes, and electric pulse protocols, resulting in different therapeutic outcomes. Shirley and colleagues (2015), compared three different electrical protocols of IT IL-12 GET (EP1: six rotating 1300 V/cm, 100 μs; EP2: ten 600 V/cm, 5 ms pulses; and EP3: one 667 V/cm, 100 ms pulse). They demonstrated that, in general, EP1 generated lower levels and a shorter expression time of IL-12 when compared to EP2, while EP3 generated similar levels to EP2, however, its analysis was discontinued after visible damage to the animals. Interestingly, they showed that the EP1 groups had similar tumor regression response to EP2 groups, however, EP1 groups showed higher immune cell infiltrate (CD11b positive cells), higher tumor rejection upon rechallenge and longer long-term survival compared to EP2 higher level expression groups, signaling the role of IL-12 in generating or abrogating an adequate immune response depends on its level and possibly duration of expression. Furthermore, they also evaluated the responses using different IL-12 plasmids, pUMVC3-mIL12 (commercially available) and pAG250-mIL12 (plasmid engineered for higher IL-12 expression), and the same inverse relation was observed, increased IL-12 correlated with reduced antitumor response in the animals. Indeed, they also demonstrated that different electrodes (plates or needles) impact gene expression levels even when using the same electric protocol [143]. Together, those findings demonstrate the importance of the correct choice of plasmids, electric protocols and electrodes for controlled gene expression. Later, the same group further evaluated the immune response unleashed by the IL-12 GET treatment with the pUCMV3-mIL-12 plasmid. They showed that responder mice had increased infiltration of immune cells in the tumor microenvironment and augmented expansion of CD4+ and CD8+ memory cells (CD44^+^) in peripheral blood, yet also showed downregulation of regulatory cells (MDSCs and CD4+ Foxp3+ Tregs) or exhausted T cells (CD4+ PD1+ and CD8+ PD1+ T cells) [157].

Tumors have an irregular and precarious vascularization when compared to normal tissues, and may also differ among different tumor types and sizes, impacting drug delivery and distribution into tumors, mainly in systemic (intravenous) administration [72,183], affecting tumor outcomes. In fact, it was shown that tumors with better vascular functions had better outcomes when compared to poorly vascularized tumors [184,185]. Further, it was demonstrated that tumors up to 3 cm^3^ had better outcomes, with a negative correlation between the tumor size and tumor outcome. Despite the lack of information, it stands to reason that in smaller lesions an IT drug administration allows a uniform drug and electric field coverage that favors EP-therapies, however, there are no data regarding IT administration in larger lesions, and systemic administration of bleomycin is indicated in these cases [42,72]. However, the aforementioned precarious vascularization is an obstacle to drug distribution, mainly in the case of large tumors which are more likely to have areas with limited blood flow, further, larger tumors also present a diminished antivascular effect of ECT, as larger vessels seem to be resistant to RE vascular disruption [72,183]. A combined approach of both IT and IV drug administration has been proposed to overcome those limitations, minimizing the necessity of further ECT sessions [183]. Even so, in larger tumors proper electric field coverage might be difficult, with a higher probability of escape by nonelectroporated cells, demonstrating the importance of a careful and well-designed therapeutic plan [72].

Furthermore, some other biological factors such as previous tumor viral infection, previous treatments, drug resistance, tumor genomics and mutational load and diverse features of tumor microenvironment and of extracellular matrix seem to impact outcomes of EP-based therapies and deserve further investigation [72].

## 7. Conclusions

Recent developments in the fields of genetic engineering and immunology research give us high expectations for improvements in the efficacy of drugs and plasmid delivery and DNA immunization using EP. In this perspective, the association of ECT and GET can be envisioned as an interesting, combined modality to generate a local response and also achieve systemic antitumor immunization [63,132]. Additionally, a complete and concise description of all material and methods used in EP-related therapies [30,72,186] and further exploration of immunological analysis aiming for reproducibility and better analytical understanding of the outcomes are suggested. Next, the evaluation of results and follow-up of clinical trials will be important for determining the success of EP-based treatments and may pave the way for personalized therapeutic protocols, assisting the advent of new therapies that improve the quality of human and animal health in the near future.

## Figures and Tables

**Figure 1 vaccines-09-00727-f001:**
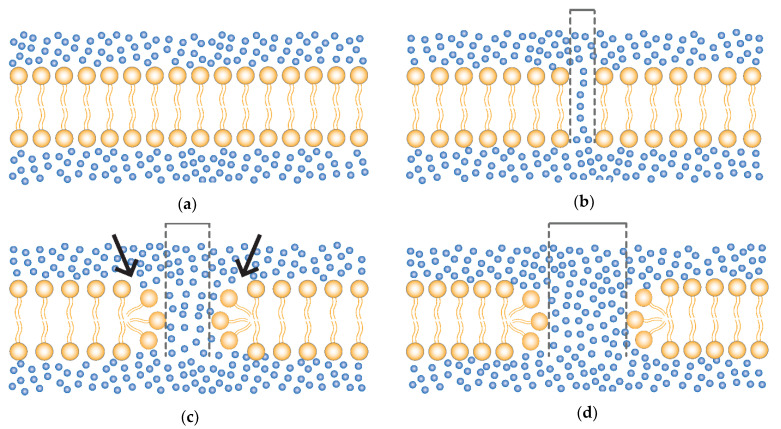
Formation of membrane pores. (**a**) Normal non-electroporated phospholipid bilayer membrane (yellow) and water (blue dots). (**b**) Early formation of a small unstable water channels. (**c**) Reorganization of the hydrophilic heads (arrow) and consequent formation of a metastable pore (reversible electroporation); with pulse cessation the membrane tends to return to the normal state (**a**); and (**d**) electric pulses beyond the RE threshold parameters leads to enlargement and destabilization of the pore with consequent loss of cell homeostasis (irreversible electroporation—IRE). Dashed lines representing the pore size.

**Figure 2 vaccines-09-00727-f002:**
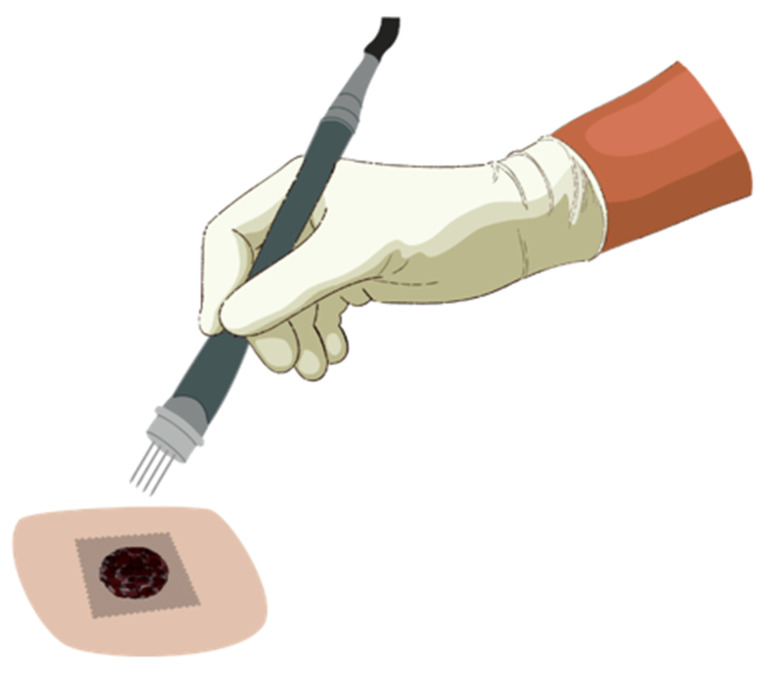
Application of electrochemotherapy in a tumor. After drug administration, the electric pulses are delivered by electrodes in contact with the tumor. The electrodes must be systematically moved to the adjacent area, and pulses applied again until the entire tumor area is covered. In the image, a model of needle electrode is exemplified.

**Table 1 vaccines-09-00727-t001:** Current human clinical trials in the recruitment phase for electrochemotherapy, electrotransfer gene and combinations. Accessed on: 1 June 2021. Link: https://clinicaltrials.gov/ct2/results?term=electroporation+NOT+irreversible+NOT+calcium+NOT+infection&recrs=a.

Research Aim	Neoplasms	Clinical Trial	Reference Centers
VGX-3100 and electroporation in treating patients with HIV positive high-grade anal lesions	Anal ntraepithelial neoplasia High grade squamous intraepithelial neoplasia and others	NCT03603808	UCLA CARE Center, United States University of California, United States and three more
Tavo with E electroporation, pembrolizumab and epacadostat in patients with unresectable head and neck cancer	Metastatic head and neck squamous cell carcinoma Recurrent head and necks quamous cell carcinoma Unresectable head and neck squamous cell carcinoma	NCT03823131	University of California, United States
Neoadjuvant immunotherapy with Tavo + electroporation in combination with Nivo. in melanoma patients	Melanoma	NCT04526730	Moffitt Cancer Center, United States
REVEAL 2 Trial (Evaluation of VGX-3100 and Electroporation for the treatment of cervical HSIL)	Cervical dysplasia Cervical high grade squamous intraepithelial lesion HSIL	NCT03721978	Visions Clinical Research, United States Western Connecticut Health Network, United States and 44 more
HPV DNA vaccine via electroporation for HPV16 positive cervical neoplasia	Human papillomavirus type 16 Cervical intraepithelial neoplasia grade II cervical intraepithelial neoplasia, grade III	NCT04131413	Johns Hopkins University, United States
Tavo and pembrolizumab in patients with stage III/IV melanoma progressing on pembrolizumab or nivolumab treatment (Keynote 695)	Stage III/IV melanoma	NCT03132675	The University of Arizona Cancer Center, United States Yuma Regional Medical Center, United States and 35 more
Tavo and pembrolizumab with or without chemotherapy in patients with inoperable locally advanced or metastatic TNBC	Triple negative breast cancer	NCT03567720	Stanford University Medical Center, United States Moffitt Cancer Center, United States and five more
Electrochemotherapy for noncurable gastric cancer	Gastric cancer	NCT04139070	Zealand University Hospital, Denmark
A study of JNJ-64300535 in healthy participants	Healthy	NCT04736147	SGS Life Science Services, Belgium.
INO 5401 vaccination in BRCA1/2 mutation carriers	BRCA1/2 mutation	NCT04367675	University of Pennsylvania, United States
DNA plasmid-encoding interleukin-12/HPV DNA plasmids therapeutic vaccine INO-3112 and durvalumab in treating patients with recurrent or metastatic human papillomavirus associated cancers	Human papillomavirus-16 positive Human papillomavirus-18 positive Metastatic malignant neoplasm and others	NCT03439085	MD Anderson Cancer Center, United States
Neoantigen DNA vaccine alone vs. neoantigen DNA vaccine plus durvalumab in triple negative breast cancer patients following standard of care therapy	Triple negative breast cancer Triple negative breast neoplasms Triple negative breast cancer and others	NCT03199040	Washington University School of Medicine, United States
A study evaluating UCART019 in patients with relapsed or refractory CD19+ leukemia and lymphoma	B cell leukemia B cell lymphoma	NCT03166878	Chinese PLA General Hospital, China
GNOS-PV02 personalized neoantigen vaccine, INO-9012 and pembrolizumab in subjects with advanced HCC	Hepatocellular carcinoma	NCT04251117	Johns Hopkins Hospital, United States Icahn School of Medicine at Mount Sinai, United States Auckland Clinical Studies, New Zealand
Safety and immune response to a mammaglobin-A DNA vaccine in breast cancer patients undergoing neoadjuvant endocrine therapy	Breast cancer Breast carcinoma Malignant neoplasm of breast	NCT02204098	Washington University School of Medicine, United States
Platform study for prostate researching translational endpoints correlated to response to inform use of novelcombinations	Metastatic castration-resistant prostate cancer	NCT03835533	Angeles Clinic, United States University of California San Francisco, United States and four more.
Electrochemotherapy of gynecological cancers	Gynecological cancers	NCT04760327	Institute of oncology Ljubljna, Slovenia
EndoVE endoscopic treatment for esophageal and gastric cancer	Esophageal cancer Esophageal disease Gastric cancer	NCT04649372	Nottingham University Hospital Nottingham, United Kingdom

Abbreviations: VGX-3100: human papillomavirus DNA plasmids therapeutic vaccine; HIV: human immunodeficiency virus; Tavo: tavokinogene telseplasmid; DNA plasmid that encodes human interleukin-12. Nivo: nivolumab. HSIL: high-grade squamous intraepithelial lesions. HPV: human papilloma virus; TNBC: triple negative breast cancer; JNJ-64300535: proposed DNA vaccine to prevent hepatitis B; INO 5401: DNA plasmid that encodes different tumor antigens; BRCA1/2: breast cancer 1 and breast cancer 2 genes, respectively; INO-3112: DNA plasmids that encodes E6 and E7 HPV related cancer antigens and human interleukin-12; UCART019: Universal CD19-specific CAR-T cell; GNOS-PV02: a personalized neoantigen DNA vaccine; INO 9012: DNA plasmid that encodes genes for human interleukin-12; HCC: hepatocellular carcinoma.

## Data Availability

No new data were created or analyzed in this study. Data sharing is not applicable to this article.

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
