# Peer review of "Clinical Applications and Immunological Aspects of Electroporation-Based Therapies"

_vaccines, 2021, doi:10.3390/vaccines9070727_

Round 1

Reviewer 1 Report

In this manuscript, the authors mainly reviewed the most recent findings about electroporation-based therapies and their application for the treatment of cancer both in humans and animals. In particular, Electrochemotherapy and Gene Electrotransfer strategies are extensively described with a focus on immunological aspects.

The manuscript is well written and summarizes interesting data for the field.

I offer the following comments in hope to assist with improvement:

1. Both Fig. 1 and Fig. 2 are missing, so in its current form the manuscript can not be evaluated in its entirety. The authors should add the figures to make a complete revision of the manuscript possible.

2. In my opinion, the only thing missing is some additional data concerning the implementation of GET protocols as many efforts are underway through in vitro and in vivo studies to optimize GET protocols to minimize tissue damage and enhance gene transfection efficiency. For example, a successful strategy to increase DNA uptake and expression is the use of hyaluronidase.

3. In Table 1, the authors can use acronyms instead of full terms and better organize the content to reduce the size of the table and make the content more immediately understandable.

Author Response

  1. Both Fig. 1 and Fig. 2 are missing, so in its current form the manuscript can not be evaluated in its entirety. The authors should add the figures to make a complete revision of the manuscript possible.
    A: Sorry for the inconvenience. We believe that the doc file lost its formatting, but the figures were certainly included in the file. We have also uploaded the figure files separately to ensure that all information is available for review.
  2. In my opinion, the only thing missing is some additional data concerning the implementation of GET protocols as many efforts are underway through in vitro and in vivo studies to optimize GET protocols to minimize tissue damage and enhance gene transfection efficiency. For example, a successful strategy to increase DNA uptake and expression is the use of hyaluronidase.
    A: We appreciate this insightful comment. Additional information was added and can be seen in lines 274-278.
  3. In Table 1, the authors can use acronyms instead of full terms and better organize the content to reduce the size of the table and make the content more immediately understandable.
    A: As suggested, we have revised the table and took advantage of the opportunity to make some additional improvements, as outlined here:
  • We have replaced the "Interventions" column with "Clinical Trial", thus providing a means for the reader to have access to the entire clinical study.
    •The number of institutions participating in the clinical study is now limited to 3 topics, with the addition of the study identifier so that the reader will be able to use the source material to find all the other institutions.
    •We chose to exclude the study “Electrochemotherapy of Posterior Resection Surface for Lowering Disease Recurrence Rate in Pancreatic Cancer”, “Neoantigen DNA Vaccine in Combination With Nivolumab/Ipilimumab and PROSTVAC in Metastatic Hormone-Sensitive Prostate Cancer” since this table only presents studies in Recruting status, which these are not.      
    • We added a new study in Recruiting status: “A Study of JNJ-64300535 in Healthy Participants”.

Reviewer 2 Report

Very nice and complete review.
comments:
line86: posterior membrane resealing.
posterior  refers to a localization, wouldn't just ' membrane resealing' be better?
line 93 : homeostatic threshold value
why is the threshold homeostatic? wouldn't just 'threshold value' be better? Is t his value tissue specific? in that case 'tissue specific threshold' would be more clear.
line 176-177. category1 overlaps with category2
line 189-190 this is a sweeping statement , please nuance or omit
line 230 and further: please define CR and PR.  are CR and PR classifications done by RECIST criteria? I presume they relate to the treated lesion only?
line 237: systemic disease progression does not indicate local ECT inefficacy. In this number of patients it does not say anything about abscopal effect either.
Table page 11-12-13: Either do not use abbreviations or explain them in a legend: nivo  tavo etc
Line 432-438: too long sentence
Line 510-514, most trials are fase1 and fase2 trials: variability is likely for a large part due to stochasticity.

Author Response

line86: posterior membrane resealing.
posterior  refers to a localization, wouldn't just ' membrane resealing' be better?
A: Corrected.

line 93 : homeostatic threshold value
why is the threshold homeostatic? wouldn't just 'threshold value' be better? Is t his value tissue specific? in that case 'tissue specific threshold' would be more clear.
A: Corrected.

line 176-177. category1 overlaps with category2
A: Corrected.

line 189-190 this is a sweeping statement , please nuance or omit
A: Corrected.

line 230 and further: please define CR and PR.  are CR and PR classifications done by RECIST criteria? I presume they relate to the treated lesion only?           
A: The definition of PR and CR and other response classifications were mentioned according to the cited articles, based on definitions of the authors, which have now been added to the text.

line 237: systemic disease progression does not indicate local ECT inefficacy. In this number of patients it does not say anything about abscopal effect either.
A: We have altered this phrase to reflect the need for studies in larger cohorts.

Table page 11-12-13: Either do not use abbreviations or explain them in a legend: nivo  tavo etc
A: The abbreviations are explained in the legend.

Line 432-438: too long sentence
A: This phrase has been altered.

Line 510-514, most trials are fase1 and fase2 trials: variability is likely for a large part due to stochasticity.  
A: We have added a phrase reflecting the insight of the Reviewer, indicating that variability may stem from the early phase trials performed with a relatively low number of patients.

Round 2

Reviewer 1 Report

The authors have addressed all of my concerns with the original manuscript. In my opinion, the revised manuscript is acceptable for publication.